# Enhancing Histological Techniques for Small Crustaceans: Evaluation of Fixation, Decalcification, and Enzymatic Digestion in *Neocaridina* Shrimp

**DOI:** 10.3390/ani15121715

**Published:** 2025-06-10

**Authors:** Rafał Karol Wild, Dobrochna Adamek-Urbańska, Artur Witold Balicki, Wiktoria Cieśla, Jakub Przybyszewski, Maciej Grzegorz Kamaszewski

**Affiliations:** 1Department of Animal Environment Biology, Institute of Animal Sciences, Warsaw University of Life Sciences (SGGW), 02-786 Warsaw, Poland; rafal_wild@sggw.edu.pl (R.K.W.); artur.balicki@me.com (A.W.B.); wiktoria_wiechetek1@sggw.edu.pl (W.C.); 2Department of Animal Breeding, Institute of Animal Sciences, Warsaw University of Life Sciences (SGGW), 02-786 Warsaw, Poland; s207142@sggw.edu.pl

**Keywords:** crustacean histology, fixation, decalcification, protocol optimization

## Abstract

The present study aimed to optimise histological techniques for small crustaceans species, Neocaridina shrimps, by evaluating different fixation, decalcification and enzymatic digestion protocols. A comparative analysis of the effects of different fixatives (neutral buffered formalin, Bouin’s fluid, Davidson’s fluid) with and without modifications such as trypsin digestion, decalcification and abdomen removal on tissue preservation, section quality and staining properties was carried out. The results showed that Davidson’s fluid provided the best fixation results, with minimal autolysis and good tissue preservation. Enzymatic digestion with trypsin increased tissue damage and autolysis, particularly in the liver and pancreas. Decalcification improved the quality of sections but increased autolysis and resulted in less specific staining. The optimal protocol involved removal of the abdomen, followed by fixation in Davidson’s fluid and decalcification, which allowed rapid penetration of the fixative, minimal autolysis and beneficial effects on the final quality of the slides. This study highlights the importance of adapting histological methods to the specific characteristics of small crustaceans, which offer an interesting alternative to vertebrate research models. Optimised techniques will improve the quality and reliability of histological analyses in crustacean research, enhancing the understanding of their biology and facilitating their use as model organisms in various scientific fields.

## 1. Introduction

Research in the broad field of biology is often based on the use of vertebrate animals, such as zebrafish (*Danio rerio*), laboratory mice (*Mus musculus*), and laboratory rats (*Rattus norvegicus*) [1,2]. Currently, a steadily increasing trend can be observed in scientific research following the principles of the 3Rs [3], using invertebrate species as research models. Crustaceans (*Crustacea*) are numerous, and their biology, especially their anatomical structure, provides insight into basic and advanced physiological and metabolic processes [4]. Due to the simpler structure of the nervous system of crustaceans compared to vertebrates, these organisms are more readily used for scientific research, allowing their distress to be minimized [5].

One of the most frequently used crustacean species is shrimp of the genus *Neocaridina*, as research using them allows not only experiments on the effects of environmental conditions [6] but also yields an understanding of the molecular mechanisms of many physiological processes [7]. Due to their short life cycle, rapid attainment of sexual maturity, and numerous offspring, shrimp may be an ideal candidate for studies in functional genomics [8] and nutritional [9], parasitological [10], toxicological [11], and genetic [12] studies. Most of the shrimp species used in research are characterized by fairly large bodies, which allow multiple types of analyses to be performed from a single individual and individual organs to be isolated. In the case of *Neocaridina* shrimp, the small body makes it difficult to conduct research in this way. Among the research methods used in experiments with *Neocaridina* shrimp, histological techniques are often used alone or as additional research methods.

Adequate preparation of materials for histological examination is key to obtaining high-quality material that allows for a range of qualitative and quantitative tissue analyses. Fixation is one of the key steps in preparing biological materials for histological examination. The most commonly used chemical fixation methods are neutral-buffered formalin, Bouin’s fluid, and Davidson’s fluid. Considering the significant species differences between the vertebrates and invertebrates used in this study, optimizing and standardizing the fixation of histological material in the latter group is necessary. Such an attempt has been made in whiteleg shrimp (*Penaeus vannamei*), paying particular attention to the presence of a highly calcified carapace, as well as to the fact that shrimp internal organs undergo autolysis shortly after death [13]. A pilot study by Bell and Lightner in 1988 [14] reported that minimal autolysis was obtained by perfusing the shrimp with Davidson’s fluid. However, this technique is only applicable to large invertebrates (greater than 10 cm in body length), in which injection is possible.

The protocol created by Bell and Lightner is considered a reference by many researchers when studying shrimp. However, some researchers have suggested increasing the fluid volume to 30% and injecting fluid into more than one site [13]. Nevertheless, owing to the small size of shrimp in the genus *Neocaridina*, it is not possible to apply this technique to fix them. As in any other tissue processing step, artifacts can occur during the fixation process, significantly impacting the results obtained from the planned analyses.

Autolysis, particularly in the digestive tract, is one of the most common artifacts in animal histology, and can be the result of several different factors, such as delaying the start of the fixation process, low rate of penetration of the fixation fluid, fixation at too high temperature, and insufficient fixation fluid [15]. Correct fixation of the material, performed immediately after animal death and under appropriate physicochemical conditions, can benefit the quality of the material. Similar problems with the quality of histological specimens also apply to other aquatic animals often used as research models, such as fish. Autolysis in fish histological samples results from extremely intensive physiological processes involving digestive enzymes and their scaly body structure, which impedes the penetration of fixative fluid. In addition to injecting the body cavity with fixative fluid, abdominal incision, decapitation, and descaling are also used to achieve faster penetration of the body cavity. These processes can also be adapted to some invertebrates. Autolysis can also have a negative impact on further processing of histological material, particularly the sectioning process. The presence of a highly calcified structure, such as a chitinous cuticle, in addition to slowing down the fixation process, can lead to sectioning problems. During this stage, it is essential that the fixed material does not contain hard tissue fragments, which can damage the microtome knives, tear the paraffin sections, and interfere with staining and subsequent microscopic analysis. However, this problem occurs not only in shrimp but also in other aquatic animals with scales or bone plates. Digestion of shrimp carapaces for histological examination has not been tested to date, although this method is used during diaphanization of large animals [16], indicating that it can also be applied to *Neocaridina* shrimp. The aim of this study was to critically evaluate current methodologies for the histological preparation of small shrimp specimens and to refine and improve the fixation protocol in order to establish a more efficient and reliable standard for future research. In particular, the study focused on resolving the problem of hard tissue fragments in the fixed material, which can cause problems during dissection, staining, and microscopic analysis of shrimp specimens.

## 2. Materials and Methods

### 2.1. Animal Housing

The tissue material used in this study consisted of whole shrimp of the genus *Neocaridina*. The parental material was cherry-red line shrimp obtained from a private breeder. To obtain sufficient individuals of similar size, shrimp were reared in three nursery tanks (thirty-three individuals each), each measuring 20 × 30 × 40 cm (24 L). Each breeding tank was equipped with a filter, a heater, and a thermostat. An approximately 7 cm-thick light substrate in the form of quartz sand and 5 g of Java moss (*Vesicularia* sp.) were placed in each aquarium. Aquaria were maintained at a constant temperature of 22 °C ± 1 °C. A photoperiod of 14 h:10 h (14 h light, 10 h dark) was used throughout the rearing period. During the 8-month (240 days) rearing period, systematic maintenance procedures were carried out: weekly water changes of approximately 20% of the water, together with filter cleaning and fortnightly mechanical removal of dead matter from the bottom of the tanks. Shrimp were fed twice a week with Tetra Crusta Menu (Tetra GmbH, Melle, Germany; composition: crude protein—45%, crude fat—9%, crude fiber—2%, moisture content—8%), which is a commercial food dedicated to freshwater invertebrates.

### 2.2. Sample Preparation

Sixty individuals of similar size (n = 60; ~15 mm, ~240 days old) were selected for histological analyses in the intermolt stage. Shrimp were euthanized by overdosing the anesthetic in an aqueous solution of tricaine methanesulfonate (MS222, Sigma-Aldrich, Saint Louis, MO, United States) at a concentration of 40 mg dm3-1. Samples were fixed with 10% neutral-buffered formalin (AQUAMED, Warsaw, Poland) (NBF), Bouin’s fluid (Diapath, Martinengo, Italy) (B), and Davidson’s fluid (165 mL 95% ethyl alcohol, 110 mL formaldehyde, 57.5 mL glacial acetic acid, 167.5 mL distilled water; CHEMPUR, Piekary Śląskie, Poland) (D).

The shrimp were collected for the fixation groups without modification (NBF, B, D), providing a baseline for subsequent modified fixation variants. To improve the cutting properties of the specimens and the quality of fixation, two types of modification performed after fixation were used: enzymatic digestion using 0.01% trypsin solution (MP Biomedicals, Solon, OH, USA) (NBFt, Bt, Dt) and decalcifying with commercial decalcifying solution (Surgipath Decalcifier II, Leica Biosystems, Nußloch, Germany) (NBFd, Bd, Dd). Due to the strong lysing effect of endogenous tissue-degrading enzymes in these variants, abdomen removal before fixation was used only in the group with decalcification (cNBFd, cBd, and cDd, respectively) (Table 1).

Following the completion of all the material preparation stages (fixation, digestion, and decalcification), the specimens were rinsed with 70% alcohol and subjected to standard histological procedures. The paraffin-embedded samples were sectioned longitudinally using a Slee CUT 5062 rotary microtome (Slee, Nieder-Olm, Germany) to a thickness of 6 μm (five slides with three sections per slide). During this process, the occurrence of artifacts was assessed based on the degree of crumbling of the material and the tendency to tear cuttings.

The slides were stained with the standard HE method using a Leica ST5010 Autostainer XL (Leica Biosystems, Wetzlar, Germany). The stained histological slides were evaluated microscopically using an NI-E microscope (Nikon, Tokyo, Japan) with a Nikon DSFi3 camera (Nikon, Tokyo, Japan) and NIS Elements AR 5.01.00 64-bit software (Nikon, Tokyo, Japan). During microscopic observations, the degree of autolysis and staining of the slides with special attention to the specificity of the staining, intensity of the colors, and the contrast of the color reaction was evaluated. Fragments of the shrimp’s hepatopancreas, abdominal striated skeletal muscles, and ventral nerve cords were evaluated for correct fixation. The method for assessing histological samples, as shown in Table 2, was based on the methodology presented by Alaeddini et al. [17] and Prasad & Donoghue [18]. Scores were summed to assess the best fixation variant.

### 2.3. Quantitative Analysis

To quantify the effect of the modified shrimp fixation techniques, cell nuclei in the longitudinal section of the muscle tissue were measured using NIS Elements AR software (Nikon, Tokyo, Japan). To evaluate the impact of the different methods of preparing samples for histological analysis, three transverse sections were selected from the muscle tissue, and 50 cell nuclei were measured from each of the selected sections.

### 2.4. Digital Image Analysis of Stainability of Tissue

Digital image analysis was performed using a script created in the SublimeText programming environment in Python language (v. 3.13.1) using OpenCV, matplotlib, pathlib, and numpy packages to determine changes in tissue coloration caused by the shrimp fixation modifications studied. For this purpose, a 300 px × 300 px photo slice was taken from each individual on which a longitudinal cross section of the muscles was visible. Images of muscle tissue in the longitudinal section were used for the analysis, because unlike other tissues, they were preserved correctly in each group and allowed for a sizable sample without visible cell nuclei. To exclude background pixels from the analysis, a mask was applied to the sections obtained to exclude pixels for which the differences in RGB values did not exceed 40. Once the background pixels were discarded, the mean RGB value was extracted from the remaining pixels along with its visualization and a color histogram showing the frequency and intensity of the colors in the image.

### 2.5. Statistical Analysis

The results of the quantitative analysis were statistically analyzed using Statistica 13 software (TIBCO Software Inc., Palo Alto, CA, USA). Normality (Shapiro–Wilk) and variance homogeneity (Levene’s) tests were performed. Statistical differences between groups were analyzed using one-way ANOVA with Fisher’s NIR post hoc test (with *p* ˂ 0.05)

## 3. Results

The modifications applied to the fixation protocols for shrimp of the genus *Neocaridina* significantly influenced the cutting process and the final quality of the specimens, which was understood as the level of fixation and overall quality of the histological specimens.

### 3.1. Unmodified Fixation

NBF-fixed specimens showed a high tendency to tear, and the material embedded in paraffin crumbled. Specimens for which Bouin’s fluid was used for fixation did not crumble, and paraffin ribbon ripping was extremely rare. A minimal tendency to rupture was observed when cutting slides fixed with Davidson’s fluid, whereas the paraffin-embedded material did not crumble.

Analysis of microscopic images of the hepatopancreas of specimens fixed without additional modifications showed varying levels of autolysis. NBF-fixed specimens were characterized by the partial preservation of the typical cellular structure at the organ’s periphery (Figure 1A1). A similar picture was observed in specimens fixed with Bouin’s fluid; however, the area of autolysis was slightly smaller (Figure 1B1). Among the shrimp fixed with Davidson’s fluid, only a minor area of the hepatopancreas was autolyzed, and most cells retained their standard structure (Figure 1C1). In addition, the most significant number of cellular details, such as chromatin and nuclei, was observed in this variant. Analysis of the fixation of the muscle fiber structure showed significant tissue shrinkage in NBF-fixed individuals (Figure 1A3). The spaces between individual muscle fibers resulting from tissue shrinkage were significantly smaller in shrimp fixed with Bouin’s fluid (Figure 1B3). Minimal shrinkage of muscle tissue was observed in specimens fixed with Davidson’s fluid (Figure 1C3). Differences in specimen fixation were also evident in neural tissue. In the specimens fixed with Bouin’s fluid (Figure 1B2), chromatin and nuclei contained within the cell nuclei were almost invisible. Further details are distinguishable in the NBF-fixed specimens (Figure 1A2), whereas the most significant level of cellular detail was obtained in the Davidson fluid-fixed specimens (Figure 1C2). In this variant, it is also possible to distinguish the boundaries between the individual neurons. The results of the scores assessed for the specimens for which fixation quality was evaluated are presented in Table 3.

The correct staining specificity of the slides was obtained for most fixation variants tested. Basophilic cellular elements were stained correctly with alum hematoxylin (purple), and acid-absorbing structures were stained with eosin (pink).

Unaltered shrimp specimens demonstrated minor differences in muscle tissue coloration. Fixation in neutral-buffered formalin (NBF) and Davidson’s fluid resulted in nearly identical muscle coloration. In contrast, fixation in Bouin’s fluid produced a more pronounced hue, as evidenced by the reduction in green pixel values.

### 3.2. Enzymatic Digestion After Fixation

In all variants in which trypsin was used, only a slight improvement in cutting parameters was found, and again in the group in which modified formalin fixation was used, the slides were most damaged.

Extensive autolysis of the hepatopancreas was observed in all variants in which trypsin digestion was used, and the best-preserved hepatopancreas structure was obtained in group Bt (Figure 2B1). Comparing the muscle structure of the trypsin-treated specimens, a slight shrinkage of the fibers was observed, with the changes being most pronounced in the NBFt group (Figure 2A3). The application of digestion did not significantly affect the size of the cell nuclei between groups; however, shrinkage of these organelles was observed compared to the no-modification groups (Table 4).

The nerve fiber structure also showed significant differences related to enzymatic digestion. In NBF-fixed specimens, trypsin treatment after fixation negatively affected the preservation of the normal structure of this tissue (Figure 2A2). The individual nerve fibers were also damaged. However, in group Bt, the visibility of nuclear details was higher, and the histological structure of the fibers was better preserved (Figure 2B2). Group Dt, however, showed a typical arrangement of nerve fibers, with the cell membranes of individual perikaryons visible. The nuclear details are visible in this group (Figure 2C2).

In all fixation methods incorporating enzymatic digestion applied post-fixation (NBFt, Bt, and Dt), the intensity of muscle tissue staining was diminished compared to the unaltered shrimp specimens. Comparable staining intensities were noted for NBF and Davidson’s fluid fixation, whereas Bouin’s fluid fixation led to a slight increase in intensity, which was observable as decreased green pixel values.

### 3.3. Decalcification After Fixation

In groups where the abdomen was not cut off, the organs were characterized by a slight occurrence of autolysis. The nerve fibers were characterized by a regular arrangement of nerve fibers, and numerous nuclear details were observed in the cell nuclei.

The hepatopancreas of individuals in the NBFd group were characterized by a smaller area covered by autolysis (Figure 3A1). In the NBFd group, muscle fibers were formed into clustered bundles rather than single fibers, with slight shrinkage of muscle fibers compared to non-treated specimens (Figure 3A3). The size of cell nuclei in the decalcified groups showed the most significant variability. There was an increase in the area of cell nuclei in the NBFd group compared to that in the Bd and Dd groups, with the increase in the Bd group being almost double that in the B group (Table 4). In the nerve tissue, the nerve fibers of individuals from the NBFd variant were characterized by the correct structure, but the level of nuclear detail obtained was low (Figure 3A2). In the Bd group, the tissue was poorly preserved compared to NBFd and Dd, in which this tissue was characterized by a regular arrangement of nerve fibers, and numerous nuclear details could be observed in the cell nuclei (Figure 3A–C2).

The specimens fixed with Davidson’s fluid, followed by decalcification (Dd), exhibited no significant differences compared to the unmodified specimens (D). Specimens fixed with NBF and Bouin’s fluid before decalcification (NBFd and Bd) showed enhanced staining intensity relative to unmodified specimens (NBF and B), as indicated by the lower blue and green channel values, with NBF-fixed specimens exhibiting the most substantial decrease (Figure 3).

### 3.4. Abdomen Removal Before Fixation

The hepatopancreas of individuals in the cNBFd group were characterized by numerous correctly fixed cells at the organ’s periphery. Still, there was significant autolysis in the center (Figure 4A1). The digestive gland subjected to the greatest autolysis was observed in the cDd group, and the level of cellular detail in the few correctly fixed cells was low, which prevented their analysis (Figure 4C1). In contrast to the cBd, and cDd groups, muscle fibers in the cNBFd group were slightly shrunken (Figure 4A-C3). There were no differences in the size of cell nuclei in the cNBFd group compared to the NBF group, whereas the other groups showed shrinkage (Table 4). As in the hepatopancreas, the most prominent degenerative changes were observed in the nervous tissue of the cNBFd group (Figure 4A2), whereas the structure of this tissue was usually preserved in the cBd and cDd groups. In the cDd group, the visibility of nuclear details in nerve cells was higher than in the cDd group (Figure 4B2,C2).

Removing the abdomen and decalcification led to a significant change in staining intensity. The green channel intensity was lower in the cNBFd group than in the NBF group. In contrast, removing the abdomen and decalcification led to an increase in green channel intensity in the cBd group compared to the B group. For shrimp with an abdomen cut before Davidson’s fluid fixation (cDd), no alteration in coloration was detected compared to other variants of the same fixative (D, Dd, Dt, tD). In the case of NBF fixation (cNBFd), cutting the abdomen did not enhance stainability compared with uncut specimens (NBFd), which maintained an intense red coloration. Conversely, Bouin’s fluid-fixed specimens (cBd) demonstrated improvement, with abdominal cutting reducing red intensity compared to uncut specimens (Bd), as indicated by the increased blue and green values (Figure 4).

## 4. Discussion

The growing interest of the academic community in moving away from vertebrate research under the 3R principle is intensifying the search for alternative animals. Invertebrates are easy to breed, readily available, and have known genomes, anatomy, and physiology [19,20,21,22,23]. The use of these animals in various fields of biology, veterinary medicine, aquaculture, and medicine has increased in recent years, leading to the necessary optimization and standardization of research methods. Fixing shrimp for histological examination can present some problems owing to the rapid autolysis of the tissues of these animals [24]. This process results from the action of proteolytic enzymes, which lead to the degradation of proteins that build tissue structures. Bell and Lightner developed a protocol [14] for the fixation of white shrimp (*Penaeus vannamei*), which is considered a reference for histological research [25,26,27]. Cerevllione et al. analyzed this protocol in 2017 [28], and proposed injecting the bodies of large white shrimp to increase the availability of fixatives to tissues that are particularly vulnerable to the onset of autolysis. However, to carry out studies under laboratory conditions, shrimp with small bodies should be used. Individuals of the genus *Neocaridina* may be ideal candidates for wide-ranging scientific research, although existing protocols are inadequate for their size and need to be optimized.

Tissue fixation is the most critical step in specimen processing for histological studies. This stage is crucial for preserving high-quality specimens that will be fit for further analysis; therefore, the choice of the appropriate fixative should be based on the type of tissue and planned analyses. The fixation process is complex, continuous, and closely related to the chemical composition of fixatives. Neutral-buffered formalin (NBF), a fixative that allows efficient fixation of small tissues, is routinely used for vertebrate histology. It is characterized by the efficient penetration of tissues of diverse histological structures, including bone tissue, but at the same time a slow cross-linking reaction rate, binding to amine groups to form methylene bridges. Compound fixatives (consisting of more than one chemical with fixed properties) have slightly different properties owing to their varied compositions. Bouin’s and Davidson’s fluids are routinely used as alternatives to NBF. Both Bouin’s and Davidson’s fluids denote fixatives that destabilize hydrophilic and hydrophobic bonds, leading to changes in the spatial conformation of proteins [29]. Formaldehyde, owing to its ability to polymerize and its particle size, has a lower rate of penetration into animal tissues (approximately 0.3–1 mm/h) than ethyl alcohol or picric acid found in the aforementioned complex fluids [30,31]. Considering the chemical structure of shrimp of the genus *Neocaridina*, it can be assumed that using a fixative fluid with a slower reaction/bond-forming time compared to others will not yield results as good as in vertebrate tissue fixation. The present study indicates that the routine procedures used in vertebrate histology are inappropriate for invertebrates with carapace.

### 4.1. Sectioning Without Modifications

The observed cutting difficulty was one of the most noticeable differences in the effects of the compared fixing fluids without modification. This is believed to be the result of the hard carapace, which is composed of chitin [32]. The tested fixative fluids did not have decalcifying properties, except for picric acid contained in Bouin’s fluid. All three fixative fluids had different modes of action on tissues, and their composition can penetrate tissues at different rates. Microscopic images of the area of tissue autolysis show that the most poorly preserved tissues were those fixed with NBF, whereas specimens fixed with Bouin’s fluid and Davidson’s fluid were fixed to a similar degree. Bouin’s and Davidson’s fluids have similar compositions, with Bouin’s fluid using picric acid instead of alcohol. Therefore, the observed differences may be the result of differences in the compositions of these fixatives and their mechanism of action. In Bouin’s fluid, picric acid forms salts with alkaline proteins, shrinking the cell nuclei; however, the addition of acetic acid coagulates the nuclear chromatin, causing swelling of this organelle, which partially nullifies this effect. In Davidson’s fluid, instead of picric acid, alcohol is used, which also has shrinking properties. As in Bouin’s fluid, acetic acid partially cancels this effect; however, judging by the results of shrinking cell nuclei, it is insufficient. Similar results were obtained when comparing Bouin’s and Davidson’s fluids in the fixation of the eye, indicating better-preserved structures of this organ in reparations from Bouin’s fluid. The authors of this study indicated that Davidson’s fluid has a higher rate of penetration and reaction with tissue than NBF [33].

Microscopic images obtained from the specimens fixed without modification showed minimal differences in terms of staining. Tissues fixed with NBF showed good staining specificity, but the intensity and contrast of the colors obtained were not as good as those in the other groups. This can be explained by formalin binding to amino groups, creating a physical blockade to negatively charged eosin molecules, resulting in reduced staining quality [34]. In Davidson’s fluid, the observed microscopic images appeared correct regarding specificity, staining intensity, and contrast. The improved quality of the slides may be due to the fact that Davidson’s fluid is characterized by a low percentage of formalin, which has little effect on the binding of dye molecules to cellular elements. In contrast, the slides obtained after fixation in Bouin’s fluid showed similar staining intensity to the samples fixed in Davidson’s fluid; however, the staining contrast was significantly lower. This staining result is a direct consequence of the fact that the fixation method using Bouin’s fluid involves both denaturing (due to the alcohol content of the formulation) and cross-link bond formation (due to the formaldehyde content of the formulation) [35].

### 4.2. Digestion with Enzymes

Enzymatic digestion commercially removes the shrimp carapace during the preparation of the raw meat [36]. In addition, enzymatic digestion is used during the preparation of biological materials for scientific research using techniques such as immunohistochemical staining [37], clearing and staining [38], detachment of cells prior to flow cytometry [39], and isolation of genetic and protein material [40]. To date, enzymatic digestion has not been used to fix biological materials for histological analysis.

The use of enzymatic digestion after shrimp fixation had little effect on the cutting process of biological material; however, significant differences were observed in the tendency of the paraffin sections to tear. As with the unmodified groups, the NBFt group also showed the greatest difficulty in obtaining uniform sections. The observed result may be due to the effect of the aqueous solution of trypsin. Water tends to partially reverse the methylene bridges formed, in a sense “reversing” the effect of formalin fixation.

The differences in machinability of the Bt and Dt specimens were due to the unique properties of these fluids. Bouin’s fluid contains picric acid, which acts aggressively on calcified tissue fragments by partially removing calcium ions from their compounds. Combined with a high penetration rate compared to NBF, such an action denaturing the fixation method does not reverse in aqueous solutions, and Bouin’s fluid achieved the best performance. These observations were also confirmed by microscopic images, which showed that the least lysed hepatopancreas was observed in the Bt group compared to the other groups. This organ is directly related to its function, which includes the production and secretion of digestive enzymes [41]. After the organism’s death, enzyme activity is not inhibited until fixation, resulting in tissue-forming proteins being degraded into amino acids and autolysis of the biological material [42]. In shrimp subjected to trypsin digestion after chemical fixation, a small area of tissue is autolyzed when the action of digestive enzymes is inhibited. This is a direct result of the supplied trypsin acting on fixed tissues, whose chemical protein structure was reinforced by additional chemical bonds formed by the action of the fixative molecules.

There were several differences in the staining of trypsin-digested specimens compared to that of non-digested specimens. NBF-fixed and trypsin-digested specimens had normal staining specificity and good color contrast, but were not intensely stained. This is due to the chemical nature of fixation with formalin, which forms cross-links, and the proteolytic effect of trypsin on the tissues. As a result of the formation of cross-links and the additional degradation of protein structures, eosin could not attach to the amino groups, resulting in reduced slide quality. In Bouin’s fluid-fixed specimens, increased color intensity was observed when enzymatic digestion was performed after fixation. It is possible that the concentration or incubation time used was insufficient to obtain satisfactory results. However, the results indicate this substance’s great potential for use in histotechnology.

### 4.3. Decalcification

The carapace contains not only chitin but also calcium, which imparts hardness to the armor. Highly calcified tissues can cause numerous problems when cutting paraffin-embedded material because of damage to knife facets [43]. Decalcifying fluids in histotechnology can be divided into three groups: strong mineral acids, weak organic acids, and chelating agents [44]. A commercial decalcifying liquid, Surgipath Decalcifier II, which is an 11% hydrochloric acid solution, was used to optimize the fixation of shrimp of the genus *Neocaridina*. Hydrochloric acid has been shown to have hydrolytic properties on chitin [45], which is the main building block of the shrimp carapace; hence, its use weakens this tough structure. Analyzing the obtained *Neocaridina* shrimp specimens, it can be concluded that decalcified specimens were more easily subjected to the cutting process than non-decalcified specimens.

In the analyzed Bd and Dd groups, a significantly larger area of autolysis was observed in the hepatopancreas than in the NBFd group. The observed changes in the hepatopancreas region were only partially due to the effects of fixatives (mentioned above) and the degenerative effects of hydrochloric acid. This may be due to the direct effect of the strong acid on animal tissue, which had previously been fixed in fixing fluids at an acidic pH. As a result of this synergic interaction (of both fixative and decalcifying solution), increased hydrolysis of delicate tissue elements may have occurred [46,47,48], resulting in the observed shrinkage of the cell nuclei and impairing the affinity of the cytoplasm for acid dyes. Microscopic image analysis of these specimens showed non-specific staining, regardless of the fixing fluid used. Due to the treatment of the tissue with hydrochloric acid, by giving the slides an acidic reaction, eosin stained the slides with an intense red color and caused non-specific staining of the cell nuclei [49].

### 4.4. Cutting off Abdomen

There are several strategies in animal histology that can accelerate infiltration of the fixation fluid, which involve dividing the material into fragments, tissue incision, perfusion, injection of fixative fluid into body cavities [14,50], or removing unnecessary tissue parts that are not necessarily intended for further study. In scientific experiments carried out on small shrimp where complete dissection is not possible, increasing infiltration of the fixative can be achieved by cutting off the abdomen [51]. Considering the results from groups without the use of modification or trypsin digestion, it can be concluded that neither technique provided sufficient results. Therefore, knowing that the specimens after decalcification were of higher quality, a decalcification step was added to the abdominal removal process to positively affect the sectioning process. This was possible because of the increased accessibility of the tissues not only to the fixative but also to the decalcifying fluid, which was also used for these shrimp fixation variants. Analysis of the autolysis of the specimens that underwent abdominal excision revealed a normal microscopic image in most slides. The degree of autolysis varied according to the fixation fluid used. The most extensive autolysis of the hepatopancreas was observed in NBF-fixed slides and minimal autolysis in Davidson’s fluid-fixed slides, which is consistent with the results of this group already discussed above. Staining of abdominal excised samples was not significantly different from staining of decalcified samples. The use of denaturing fixatives affected the apparently highest level of nuclei shrinkage.

## 5. Conclusions

Based on the results obtained, several conclusions regarding fixation techniques for *Neocaridina* shrimp specimens can be drawn. A fixative composed solely of formalin is inadequate for preserving the internal organs of shrimp because of the reaction rate of formaldehyde with the amino groups. As corroborated by other scientific studies, Davidson’s fluid generally yields the most favorable fixation outcomes, with minimal autolysis, excellent tissue preservation, and optimal staining intensity and contrast. Enzymatic digestion with trypsin increases tissue damage and autolysis, particularly in the hepatopancreas. Decalcification following fixation enhances sectioning quality, but exacerbates tissue autolysis and causes non-specific staining. Removing the abdomen before fixation and decalcification produced the best overall results, facilitating rapid fixative penetration, minimizing autolysis, and maintaining good staining properties. The optimal fixation protocol for *Neocaridina* shrimp involves removal of the abdomen, followed by fixation with Davidson’s fluid and subsequent decalcification, effectively balancing tissue preservation, ease of sectioning, and staining quality. This study highlights the importance of considering tissue-specific characteristics when developing histological methods for crustaceans. Future studies should investigate the application of this protocol to other small crustacean species, potentially enhancing histological analyses across a broader range of aquatic organisms.

## Figures and Tables

**Figure 1 animals-15-01715-f001:**
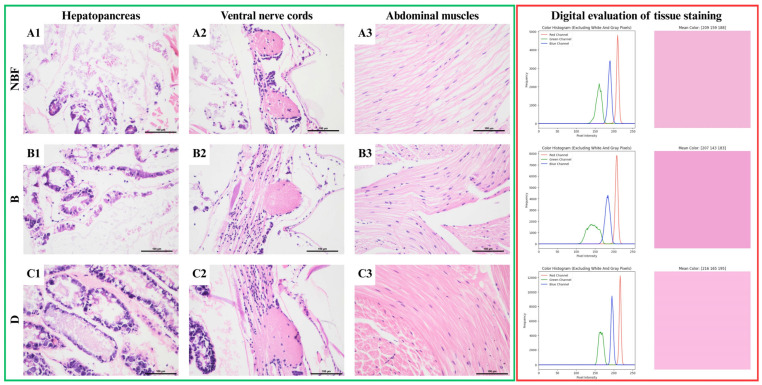
Shrimp tissue comparison for unmodified fixation variants. Green frame: microscopic images of: (**A1**–**C1**)—hepatopancreas, (**A2**–**C2**)—nervous tissue, (**A3**–**C3**)—muscle tissue. HE stain, scale 100 μm. Red frame: digital image analysis of stainability of tissue (raw image, image with mask applied, color histogram of image, and mean RGB value of image as color).

**Figure 2 animals-15-01715-f002:**
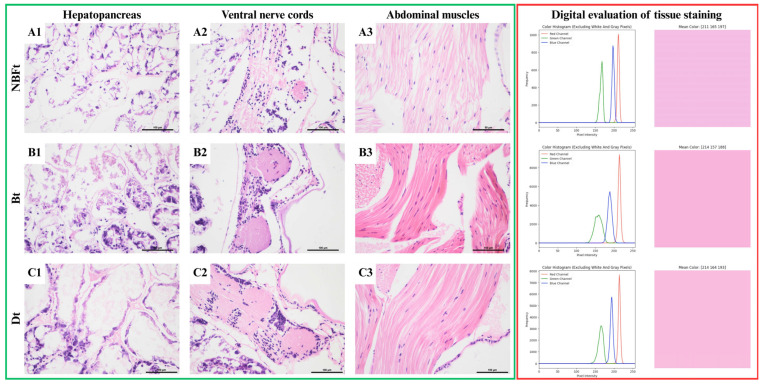
Shrimp tissue comparison for enzymatic digestion fixation variants. Green frame: microscopic images of: (**A1**–**C1**)—hepatopancreas, (**A2**–**C2**)—nervous tissue, (**A3**–**C3**)—muscle tissue. HE stain, scale 100 μm. Red frame: digital image analysis of stainability of tissue (raw image, image with mask applied, color histogram of image, and mean RGB value of image as color).

**Figure 3 animals-15-01715-f003:**
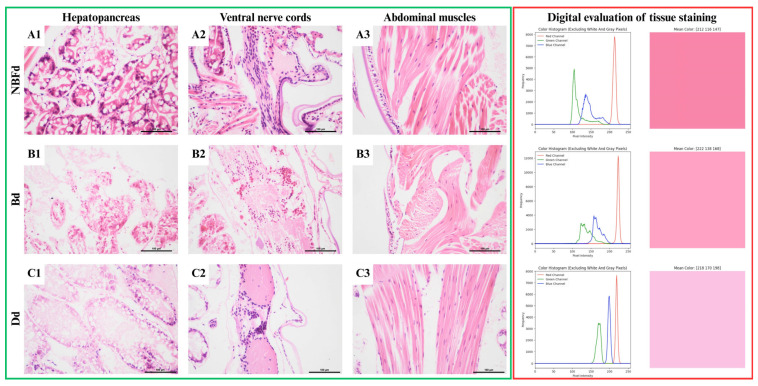
Shrimp tissue comparison for decalcification fixation variants. Green frame: microscopic images of: (**A1**–**C1**)—hepatopancreas, (**A2**–**C2**)—nervous tissue, (**A3**–**C3**)—muscle tissue. HE stain, scale 100 μm. Red frame: digital image analysis of stainability of tissue (raw image, image with mask applied, color histogram of image, and mean RGB value of image as color).

**Figure 4 animals-15-01715-f004:**
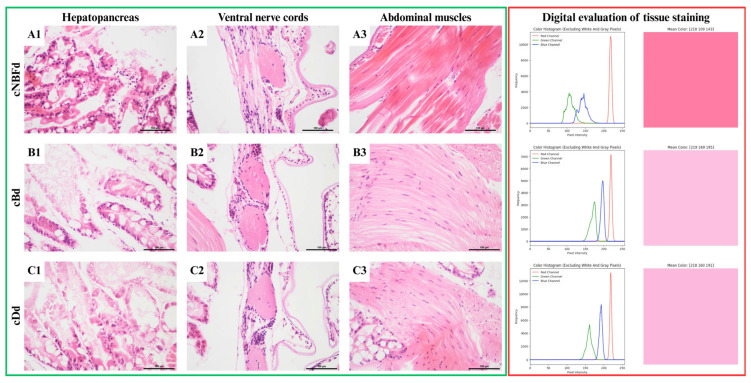
Shrimp tissue comparison for abdomen removal fixation variants. Green frame: microscopic images of: (**A1**–**C1**)—hepatopancreas, (**A2**–**C2**)—nervous tissue, (**A3**–**C3**)—muscle tissue. HE stain, scale 100 μm. Red frame: digital image analysis of stainability of tissue (raw image, image with mask applied, color histogram of image, and mean RGB value of image as color).

**Table 1 animals-15-01715-t001:** Applied variants of modification for preparing shrimp for further processing steps, along with the time intervals used.

Variant	Group	Abdomen Removal	Liquid I	Time	Liquid II	Time
No modifications	NBF	No	NBF	16 h	-	-
	B	No	Bouin’s fluid	16 h	-	-
	D	No	Davidson’s fluid	16 h	-	-
Digestion	NBFt	No	NBF	16 h	Trypsin	8 h
	Bt	No	Bouin’s fluid	16 h	Trypsin	8 h
	Dt	No	Davidson’s fluid	16 h	Trypsin	8 h
Decalcification	aNBFd	No	NBF	24 h	Decalcifier	24 h
	Bd	No	Bouin’s fluid	24 h	Decalcifier	24 h
	Dd	No	Davidson’s fluid	24 h	Decalcifier	24 h
	cNBFd	Yes	NBF	24 h	Decalcifier	24 h
	cBd	Yes	Bouin’s fluid	24 h	Decalcifier	24 h
	cDd	Yes	Davidson’s fluid	24 h	Decalcifier	24 h

**Table 2 animals-15-01715-t002:** Evaluation criteria and scoring system for analyzing the quality of fixation and staining of samples during sectioning and after staining on the hepatopancreas, muscle, and nervous tissue.

Evaluated Parameter	Score			
	0	1	2	3
Fragility and tendency to tear	High	Average	Low	Minimal
Rate of autolysis	High	Medium	Low	None
Specificity of staining	None	Low	Average	High
Staining intensity	None	Low	Average	High
Contrast of staining	Very poor	Poor	Average	High

**Table 3 animals-15-01715-t003:** Mean scores assessed for each fixation variant to evaluate the quality of sectioning, fixation, and staining of the samples.

Group	Ms	MF	MS	MI	MC	MA	MH	MM	MN	FS
**NBF**	2.6	1.2	1.8	0.4	0.4	1	1	0.4	1	**9.8**
**B**	2.8	2	2	2	1.2	2	1.8	2	2.8	**18.6**
**D**	2.8	2.6	2.6	2	2.2	2.4	2	2.6	2.2	**21.4**
**NBFt**	0.8	0.6	2.4	1.2	2	0.2	0.2	1.6	2.8	**11.8**
**Bt**	0.8	0.2	2	2.8	2	0.6	1	2.2	2.8	**14.4**
**Dt**	2.2	2.2	2	1.8	2.4	2.4	1	2.2	2.4	**18.6**
**NBFd**	2.6	1.4	0.6	2.8	0.2	0.6	0.4	0.2	1.6	**10.4**
**Bd**	2.8	2.6	0.4	2.8	1.2	0.2	1	1	2.2	**14.2**
**Dd**	2.8	2	1.4	2.8	1.2	0.6	2.2	2.2	2	**17.2**
**cNBFd**	2.6	2.2	1.2	2.2	1.6	1	0.8	2	2.8	**16.4**
**cBd**	2.6	3	0.2	2.8	0.2	2.2	2.2	2.4	2.6	**18.2**
**cDd**	3	3	1	2.8	0.8	2.8	2.8	3	2.8	**22**

Ms, mean score of specimen sectioning; MF, mean score of specimen fragility; MS, mean score of stain specificity; MI, mean score of stain; MC, mean score of stain contrast; MA, mean score of autolysis rate; MH, mean score of hepatopancreas fixation; MM, mean score of muscle tissue fixation; MN, mean score of nerve tissue fixation; FS, final score.

**Table 4 animals-15-01715-t004:** Statistical analysis of cell nucleus shrinkage in muscle tissue expressed as the mean area of the cell nuclei (µm^2^). Statistical significance was evaluated in groups of fixatives rather than among all tested fixation variants.

Fixative	Unmodified		Digestion		Decalcification		Abdomen removal	
	Mean	SD	Mean	SD	Mean	SD	Mean	SD
**NBF**	57.94 ^b^	7.31	48.27 ^a^	5.08	57.99 ^b^	11.22	58.79 ^b^	14.76
**Bouin’s fluid**	47.83 ^d^	5.55	51.26 ^c^	7.61	26.88 ^a^	12.74	44.46 ^b^	7.71
**Davidson’s fluid**	58.24 ^d^	9.49	50.41 ^c^	11.95	45.95 ^a^	12.40	47.54 ^b^	11.26

Means ± SD; n = 5; different letters indicate statistically significant differences between groups (*p* ˂ 0.05).

## Data Availability

Data are contained within the article.

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
