# Peer review of "Enhancing Histological Techniques for Small Crustaceans: Evaluation of Fixation, Decalcification, and Enzymatic Digestion in Neocaridina Shrimp"

_animals, 2025, doi:10.3390/ani15121715_

Round 1
Reviewer 1 Report
Comments and Suggestions for Authors
They are added in word file attached

Author Response
We are very thankful to the reviewers for their thoughtful and reliable reviews. We revised our manuscript in the light of the suggestions and comments. We hope our revision has improved the paper satisfactorily. Each reviewer’s comment is first recalled (in italics), then the corresponding replies are given.
Abstract
Line 13-14. Substitute Neutrally buffered formalin by neutral-buffered formalin. The percentage is missing but I suppose is 10%.
The neutrally buffered formalin was corrected to 10% neutral-buffered formalin.
Keyword
I suggest using crustacean histology rather than invertebrate histology.
The keyword: “invertebrate histology” was corrected to “crustacean histology”.
Introduction
Line 47-50. “However,…methods”. I suggest to change it since it does not make much sense.
This part was corrected to clarify more precise the sense.
Line 55. Delete ”species”.
This sentences was rewritten.
Line 73. “..high a temperature…”. Delete “a”.
“A” was deleted.
Materials and Methods
Line 118. Substitute Neutralised buffered formalin by neutral-buffered formalin. The percentage is missing but I suppose is 10%.
The neutrally buffered formalin was corrected to 10% neutral-buffered formalin.
You do not mention how many slides per treatment and number of section per slide are analysed. I think this issue deserves to be better explained.
Thank you for this comment. We indeed did not include this information, it did not seem methodologically relevant to us, for which we apologise. Five slides were prepared for each individual with three scrapings on each slide. This information has been added to the material and methods section.
Line 150. Use “Abdominal striated skeletal muscle” instead of “Abdominal muscles”.
The “abdominal muscle” was corrected to “abdominal striated skeletal muscle”.
Line 163. To evaluate the impact of the fixation options or to evaluate the impact of the different histological techniques employed?
I'm not certain whether you are referring to the style or the meaning of the sentence? Analyses of the surface of the cell nuclei of the muscle tissue have been used to quantitatively analyse the modifications used in shrimp fixation, which are part of procedures using histological techniques.
Statistical analysis
P values is missing.
The P value (P˂0, 05) was added.
Results
There are a lot of calls throughout the result section to observed different effects (autolysis, cell nuclei and muscle fiber shrinkages,….) produced by the different histological techniques employed, but the magnification of the figures as well as the size of the micrographs are small so the descriptions done cannot be ascertained. I suggest to use larger micrographs together with the use of inserts to highlight the abovementioned effects produced by the different histological techniques employed.
The photo panels included in the text have been prepared in high resolution. They consist of large images, which required compression to be able to include them in the file. Possibly there was a reduction in their quality during this process for which I sincerely apologise. In returning the file after review, I will include the photo panels as separate files to provide access to the high-resolution images.
Line 285. What do you mean with “Nerve pili”?
According to the English dictionary I used, this is a translation from the original language referring to the neural tissue weaving, however, the term is not quite correct in this context hence it has been corrected to “this tissue”.
Line 303. Liver pancreas. I suggest using a different term such digestive gland, midgut gland, gastric gland, etc.
The “liver pancreas” was corrected to “digestive gland”.
Discussion
Line 341. “The Cerevllione team…”. I suggest using “ Cervellione et al. analysed…”.
“The Cerevllione team” was corrected to “ Cervellione et al.”.
Line 344. Use “Ben and Lighter’s protocol [14] instead of “The Bell and Lighter [14] protocol…”.
The number of reference was moved.
Line 455-456. I suggest to rewrite it.
The sentence was rewritten.
Line 461-462. “….significantly shrinking the cell nuclei and increasing the affinity of the cytoplasm to the tissue.” It seems like something is missing or wrong since the sentence cannot be completely understood.
The sentence was rewritten.
References
References 1, 3, 14, 29, 31, 50 and 51 seem to be incomplete.
Online references 34 and 44 are incomplete. Please add the url.
The references were re-checked and supplemented.
Reviewer 2 Report
Comments and Suggestions for Authors
This paper is an evaluation of histological techniques for shrimp. Shrimp are important for several applications, yet evaluations of techniques is very limited. This paper was generally well written and appears a thorough evaluation of approaches for tissue analyses. There are a series of minor corrections the authors should consider and one major question they should probably address.
Major – The crustacean molting cycle is an ongoing process and should be considered in this presentation. Perhaps simply stating all shrimp were in an intermolt phase would be sufficient as it is probably unlikely the authors can discern that after the fact. However, they may want to consider how their findings might change if shrimp were in different stages off molting.
Minor
Scientific names are commonly italicized in technical papers. Most were not in this manuscript including lines 32-33, 40, 58, 67, 91, 99, 104, 189, 340, 348, 366, 450, 452, 488, 499.
Use of the word “showed” (line 61) is increasing in the technical literature, when “reported” would be better choice. Data never showed anyone anything.
Line 102, 118 – dm3? Does this refer to cubic decameters? This is an atypical presentation of volume and one likely to confuse readers.
Line 125 – colon, should be semi-colon
Table 1 – “tripsin” was misspelled.
Accepted level of probability is commonly presented in the Methods section, yet missing in this manuscript.
Line 211 – Insert “In the” in front of “Specimens fixed……..”
Line 303 – “liver pancreas” was used here, yet hepatopancreas was used later in the same paragraph and throughout the paper.
Several citations are incomplete. Lack of publishers names, location, etc.
Author Response
We are very thankful to the reviewers for their thoughtful and reliable reviews. We revised our manuscript in the light of the suggestions and comments. We hope our revision has improved the paper satisfactorily. Each reviewer’s comment is first recalled (in italics), then the corresponding replies are given.
This paper is an evaluation of histological techniques for shrimp. Shrimp are important for several applications, yet evaluations of techniques is very limited. This paper was generally well written and appears a thorough evaluation of approaches for tissue analyses. There are a series of minor corrections the authors should consider and one major question they should probably address.
Major – The crustacean molting cycle is an ongoing process and should be considered in this presentation. Perhaps simply stating all shrimp were in an intermolt phase would be sufficient as it is probably unlikely the authors can discern that after the fact. However, they may want to consider how their findings might change if shrimp were in different stages off molting.
Thank you for your valuable comment on the crustacean moulting cycle. We agree that this is an important aspect and we recognise this as a limitation of our study. We hope to be able to repeat our study on a larger number of individuals taking into account both the stage of molting and sex, which we think are very interesting parameters to verify. Nevertheless, in our study on a smaller number of individuals we could not take these parameters into account.
In response to the reviewer's suggestion, we propose to add the following statement in the methodology section (line 122): "… in the intermolt stage”.
Minor
Scientific names are commonly italicized in technical papers. Most were not in this manuscript including lines 32-33, 40, 58, 67, 91, 99, 104, 189, 340, 348, 366, 450, 452, 488, 499.
The text has been re-checked and changes made.
Use of the word “showed” (line 61) is increasing in the technical literature, when “reported” would be better choice. Data never showed anyone anything.
The sentence was corrected.
Line 102, 118 – dm3? Does this refer to cubic decameters? This is an atypical presentation of volume and one likely to confuse readers.
The unit was corrected.
Line 125 – colon, should be semi-colon
In line 125, we mention two options for fixation - we have therefore used a colon. Is it certain that we should correct it with a semicolon?
“To improve the cutting properties of the specimens and the quality of fixation, two types of modifications performed after fixation were used: enzymatic digestion using 0.01% trypsin solution (MP Biomedicals, Solon, USA) (NBFt, Bt, Dt) and decalcifying with commercial decalcifying solution (Surgipath Decalcifier II, Leica Biosystems, Nußloch, Germany) (NBFd, Bd, Dd).”
Table 1 – “tripsin” was misspelled.
The name of the enzyme name has been corrected
Accepted level of probability is commonly presented in the Methods section, yet missing in this manuscript.
The P value was added to Material and method section (line 197).
Line 211 – Insert “In the” in front of “Specimens fixed……..”
The “In the” was added.
Line 303 – “liver pancreas” was used here, yet hepatopancreas was used later in the same paragraph and throughout the paper.
The “liver pancreas” was corrected to digestive gland.
Several citations are incomplete. Lack of publishers names, location, etc.
The references were re-checked and supplemented.
Round 2
Reviewer 1 Report
Comments and Suggestions for Authors
See the attached file

Author Response
Dear reviewer,
We revised our manuscript in light of your suggestions and comments. We hope our revision has improved the paper satisfactorily.
Line 170-174. Regarding the histological techniques term, perhaps I did not make myself clear.
The thing is that you are not only evaluating the effect of fixation but also the effect of other techniques such as trypin and decalcification, that is why all together are mentioned as histological techniques.
Thank you for the clarification. We agree that this passage needs clarification, so we have corrected it to “different methods of preparing samples for histological analysis”.
Line 285. Perhaps the correct term was not nerve pili but neuropil or neuropile.
I have encountered various forms of translation of this word - hence we have reworded the sentence to avoid consternation.
Line 351. Remove “The”.
“The” was removed.
Line 354-359. I suggest rewriting it since it is not well written in my opinion.
This part was rewritten.
Line 360-361. I suggest the change of “Fixation of materials for histological studies is the most critical step in specimen processing” by “Tissue fixation is the most critical step in specimen processing for histological studies”.
The sentence was change according to your suggestion.
Line 470. Correct “Overplapping” by “overlapping” but I suggest using instead synergic interaction or synergy.
We change the “overplapping” to “syngergic interaction”.
Line 474. Delete “to”.
The first part of this paragraph was rewritten.
Kind regards,
Authors